# Leveraging Public Data to Predict Global Niches and Distributions of Rhizostome Jellyfishes

**DOI:** 10.3390/ani13101591

**Published:** 2023-05-09

**Authors:** Colin Jeffrey Anthony, Kei Chloe Tan, Kylie Anne Pitt, Bastian Bentlage, Cheryl Lewis Ames

**Affiliations:** 1Marine Laboratory, University of Guam, Mangilao, GU 96923, USA; bentlageb@triton.uog.edu; 2Graduate School of Agricultural Sciences, Tohoku University, Sendai 980-8572, Japan; 3Faculty of Agriculture, Tohoku University, Sendai 980-8572, Japan; tan.kei.chloe.p2@dc.tohoku.ac.jp; 4Coastal and Marine Research Centre, Griffith Institute for Tourism Research, School of Environment and Science, Gold Coast Campus, Griffith University, Southport, QLD 4222, Australia; k.pitt@griffith.edu.au; 5Coastal and Marine Research Centre, Australian Rivers Institute, School of Environment and Science, Gold Coast Campus, Griffith University, Southport, QLD 4222, Australia

**Keywords:** macroecology, marine ecology, niche modeling, distribution modeling, random forests model, online databases, citizen science, iNaturalist, Cnidaria, Scyphozoa

## Abstract

**Simple Summary:**

With human activities and climate change threatening biodiversity, marine resource managers must establish globally oriented, data-driven conservation practices. As the internet expands and the world becomes more connected, science is more accessible than ever, requiring only the internet to access powerful computing tools and expansive databases. Here, we demonstrate the power of citizen science, online databases, and open-source tools by using citizen-derived jellyfish reports from iNaturalist.org (accessed on 3 November 2022) in conjunction with publicly available environmental data to predict the distribution of the most conspicuous and economically relevant group of marine jellyfishes (Rhizostomeae). Online databases come with many biases, most of which can be tied back to resolution. The integration of distribution data from the published literature allows us to evaluate citizen-derived data quality and make a plan for improving data resolution. Going forward, expanding collaborations and citizen participation in underrepresented regions will decrease participation biases and improve data resolution, increasing the power of online databases and their potential to inform marine management strategies.

**Abstract:**

As climate change progresses rapidly, biodiversity declines, and ecosystems shift, it is becoming increasingly difficult to document dynamic populations, track fluctuations, and predict responses to climate change. Concurrently, publicly available databases and tools are improving scientific accessibility, increasing collaboration, and generating more data than ever before. One of the most successful projects is iNaturalist, an AI-driven social network doubling as a public database designed to allow citizen scientists to report personal biodiversity reports with accuracy. iNaturalist is especially useful for the research of rare, dangerous, and charismatic organisms, but requires better integration into the marine system. Despite their abundance and ecological relevance, there are few long-term, high-sample datasets for jellyfish, which makes management difficult. To provide some high-sample datasets and demonstrate the utility of publicly collected data, we synthesized two global datasets for ten genera of jellyfishes in the order Rhizostomeae containing 8412 curated datapoints from both iNaturalist (*n* = 7807) and the published literature (*n* = 605). We then used these reports in conjunction with publicly available environmental data to predict global niche partitioning and distributions. Initial niche models inferred that only two of ten genera have distinct niche spaces; however, the application of machine learning-based random forest models suggests genus-specific variation in the relevance of abiotic environmental variables used to predict jellyfish occurrence. Our approach to incorporating reports from the literature with iNaturalist data helped evaluate the quality of the models and, more importantly, the quality of the underlying data. We find that free, accessible online data is valuable, yet subject to biases through limited taxonomic, geographic, and environmental resolution. To improve data resolution, and in turn its informative power, we recommend increasing global participation through collaboration with experts, public figures, and hobbyists in underrepresented regions capable of implementing regionally coordinated projects.

## 1. Introduction

With human activities and climate change threatening biodiversity, establishing global monitoring techniques is a priority to improve data-driven management and conservation practices. As the internet expands and the globe becomes more connected, science is more accessible than ever via free, powerful computing tools and public databases available to those with access to the internet. Citizen science is beneficial to both communities and scientists by providing the potential for anyone to contribute to scientific knowledge, which, if utilized, facilitates environmental stewardship and increased scientific literacy while simultaneously generating large, publicly available datasets [1,2,3]. One of the most popular projects is iNaturalist (inaturalist.org), a social platform doubling as a public database that uses a combination of computer vision, automated reasoning, and machine learning to increase the quality of biodiversity reports. This platform allows non-experts to report expert-level data [4], cultivating a community of passionate citizen scientists who have provided images documenting over 125 million wildlife reports as of January 2023. iNaturalist has proven useful for conservation and management by inferring distributions of rare, dangerous, and charismatic organisms [5,6,7,8].

The ‘true jellyfish’ Scyphozoa have complex life cycles, typically characterized by a planula, polyp, and medusa stage; however, they are most readily recognized by their swimming medusa stage [9] that can cause harmful stings. Jellyfish blooms generally occur when environmental conditions promote species-specific metamorphosis of the sessile ‘polyp’ form into a tiny swimming ‘ephyra’ form, and eventually a sexually mature ‘medusa’ form [10]. The resulting phenomenon is the perception of a sudden increase in biomass of a particular jellyfish species in coastal waters or in fishing grounds, resulting in increased sting incidents for beachgoers and fishers [11,12,13,14]. Jellyfish (order Rhizostomeae) are important components of the marine trophic structure [15,16,17], harbor biomedically relevant compounds [13,18,19], sting humans [20], threaten fisheries [21,22], and are an important human food source [23,24,25]. Despite the ecological and economic importance of jellyfish, compared to marine vertebrates they are disproportionately understudied due to limited funding opportunities, ephemeral and patchy population dynamics, and cryptic species boundaries [26]. Multiple large scale synthesis studies have been conducted to better understand jellyfish macroecology [27,28,29], each with their own strengths. As biodiversity dips drastically and unequally across species and geographic boundaries, we must continue to explore and develop methods capable of tracking global patterns of all marine species, jellyfish included. This would be used to lay the groundwork for hypothesis-driven research and management to better predict jellyfish blooms, invasions, and range expansions. The ideal database and supporting analyses will establish baseline diversity and abundance metrics which can be followed over the short- and long-term in the Anthropocene.

To explore research options capable of evaluating online databases within a global context, we selected jellyfish of the taxonomic order Rhizostomeae. Their conspicuous colors, relatively large sizes, and common beachings make them popular photographic subjects for citizen contributions to iNaturalist. Herein, we use a global dataset of citizen-derived, rhizostome jellyfish reports from iNaturalist in conjunction with distribution data gleaned from the last 120 years of scientific literature and online environmental databases to provide a preliminary framework for integrating publicly available data into modeling the global niche space and predicting suitable environments of marine organisms.

## 2. Materials and Methods

### 2.1. Data Curation

Metadata (date, geographic coordinates, taxonomic information) associated with ‘Research grade’ reports for the taxonomic order Rhizostomeae Cuvier [30] were exported from iNaturalist.org using the site’s API (application programming interface) on 3 November 2022 [31]. Sample-size-based rarefaction and extrapolation curves were computed to determine the sampling coverage based on coordinate diversity (number of unique geographic coordinates) and report number for each genus using the iNEXT package v3.0.0 [32]. All iNaturalist reports were treated as genus-level reports, given that thorough morphological and molecular analyses are required for species-level resolution in rhizostome jellyfishes [33,34,35,36]. Prior to curve modeling, raw report data were converted to abundance data for each geographic coordinate by summarizing all jellyfish reports to the count of reports for that genus from that geographic coordinate rounded to the tenths digit. Of the genera identified, only those with a rarefied sampling coverage (SC) greater than 95% (i.e., saturated report data) were retained for further analyses.

To pair reports (i.e., rhizostome jellyfish photographs) in iNaturalist with publicly available marine environmental data, global monthly means for temperature [37], salinity [38], dissolved oxygen, percent oxygen saturation [39], silicate, phosphate, and nitrate [40] at 0, 5, 10, and 20 m depths were acquired from the NOAA world ocean atlas (WOA18) [41,42]. To prevent data scarcity and accommodate for land-based jellyfish reports (e.g., beachings), any environmental data within two degrees latitude/longitude and the same month as the jellyfish report were paired with the jellyfish report. Environmental data paired to the same month as the iNaturalist report account for jellyfish populations’ seasonal specificity. Multiple possible environmental conditions were often assigned to a single report because multiple environmental profiles would fulfill the matching criteria. All environmental conditions that successfully paired with a jellyfish report were subsequently averaged for each unique iNaturalist report, creating a representative, paired environmental profile given the environmental data and creating a representative dataframe hereafter referred to as the ‘curated multivariate abundance and environmental dataframe’. All data curation and statistical analyses were completed using packages built for R v4.1.2 [43] in RStudio v2022.07.2 [44], a widely used, open-source computing language. Curation was primarily handled using tidyverse v1.3.2 package collection [45].

A second distribution dataset was generated from previous publications (legacy distribution data) to evaluate iNaturalist reports and modeled distributions. Legacy distribution data included the genus and species name, its region, and the coordinates associated with the jellyfish report. Taxonomy was verified post-hoc to make sure all included members were within the correct genus and species name. Reports were removed if the taxonomic reassignment was not to one of the ten genera included in this study.

### 2.2. Statistical Models

Modeled from the curated multivariate abundance and environmental dataframe, niche space was inferred by multivariate generalized linear models (GLM) using the ‘manyglm’ function with standard parameterization as integrated in mvabund v4.2.1 [46]. This model was selected to reduce confounding effects of location and dispersion typically associated with other analyses used for high-dimension data exhibiting low-abundance means [47,48,49]. GLMs were inferred with both 1° and 2° latitudinal and longitudinal mapping to verify that our selected coordinate scale did not influence data interpretation.

Following this visual quality check, in an attempt to identify the most informative environmental variables (i.e., strong influence), random forest models (RFMs) were trained against a random selection of 80% of the data in the mapped jellyfish-environment dataframe to identify the most informative environmental variables using a combination of packages: randomForest v4.7.1.1 [50], caret v6.0.93 [51], e1071 v1.7.12 [52], and caTools v1.18.2 [53]. This also included depth-specific measurements (0, 5, 10, 20 m) for each environmental predictor from the WOA18 online database [41,42]. Environmental profiles with report data across any genus were maintained under the assumption that one or more positive reports indicates negative reports for all other genera, as it can be assumed that if an observer documents one jellyfish, they are likely to photograph and report any other jellyfish in the area. The remaining 20% of the data in the mapped dataframe was used to test the RFM accuracy and specificity.

Machine learning algorithms are a type of AI that use statistical techniques to identify patterns in data and make predictions or decisions. They can handle complex dataframes by automatically identifying relevant features and patterns, which are well suited for tasks such as image recognition, natural language processing, and predictive modeling. Therefore, machine learning algorithms can fit more complex models than GLMs and provide additional inferred niche resolution [54]. Additionally, the flexibility associated with machine learning algorithms makes is easier to fulfill model assumptions [54,55], especially compared to the typically violated linear assumption for GLMs [56]. RFMs are one approach that combines machine learning with binary decision trees and can be used to infer ecological niches based on a set of predictors [55]; therefore, this is the method we use here to predict jellyfish occurrence based on abiotic environmental factors.

To model the most suitable habitats for each genus, geographic coordinates were retained if their environmental data fell within the inner 90th percentile of the RFM’s best predictors’ environmental distributions generated from positive report data (inferred environmental space). For example, according to its RFM, the most suitable habitat for *Lychnorhiza* was inferred based on the distribution built from positive reports for the three most informative variables: silicate at 10 m, phosphate at 20 m, and salinity at 0 m. The threshold for variable inclusion was determined by the largest difference in ‘mean decrease Gini’ (herein referred to as ‘relative importance’) between two ordered variables. In cases where the first largest difference only included one variable, the rule was extended to the second largest difference. Datasets comprising the predicted suitable environmental range (inner 90th percentile) were mapped to a global map as a two-dimensional density distribution using ggplot2 v3.4.0 [57].

In addition to the RFM integrated accuracy evaluation, distribution predictions were evaluated for accuracy by comparing predicted distributions to occurrence reports from legacy distribution data (Appendix A). Distributions for a particular genus were considered accurately predicted when RFM modeled distribution overlapped with reports from legacy data two-dimensional density distribution. Accuracy was calculated as the number of legacy reports within the predicted distribution divided by total legacy reports within that genus multiplied by 100.

All original code and raw data files have been deposited into GitHub and are publicly available (https://doi.org/10.5281/zenodo.7897878).

## 3. Results

### 3.1. Taxonomic Representation and Distributions of iNaturalist Reports

Ten genera within the order Rhizostomeae Cuvier [30] were determined to have saturated report data from iNaturalist (SC > 0.95), as informed by rarefaction: (1) *Cassiopea* (*n* = 789), (2) *Catostylus* (*n* = 788), (3) *Cotylorhiza* (*n* = 946), (4) *Eupilema* (*n* = 40), (5) *Lobonema* (*n* = 20), (6) *Lychnorhiza* (*n* = 98), (7) *Pseudorhiza* (*n* = 202), (8) *Rhizostoma* (*n* = 1819), (9) *Rhopilema* (*n* = 423), (10) *Stomolophus* (*n* = 2682) (Figure 1 and Appendix A).

Reports from iNaturalist (photographs) primarily come from North America, Europe, and Australia, with fewer reports coming from South America, Africa, and Asia (Figure 1A). In general, modeled latitudinal (X^2^ = 5911.3, *p* < 0.001) and longitudinal (X^2^ = 5175.7, *p* < 0.001) distributions are highly genus-specific, forming seven statistical groups (a–f) comprising the ten genera (Figure 1). *Rhizostoma* reports push far into Northern Europe, in addition to overlapping with the more southern, Mediterranean genus *Cotylorhiza*. Alternatively, some groups have very similar latitudinal distributions including *Eupilema*, *Lobonema*, *Lychnorhiza*, and *Pseudorhiza* (group d), or *Stomolophus* and *Rhopilema* (group f) (Figure 1B).

### 3.2. Niche Modeling and Distribution Predictions

As suggested by the iNaturalist reports (Figure 1), environmentally informed niche modeling supported three zones of niche space inhabited separately by (1) *Stomolophus*, (2) *Rhizostoma*, and (3) all other genera (Figure 2A). The coordinate range used to pair report and environmental variables (1° vs. 2°) did not change the niche model interpretation (Appendix A), validating the use of a coarser resolution (2°) to maximize jellyfish report sample size.

Despite highly overlapping niche spaces, genus distribution was often predicted best by different environmental variables, as inferred by relative importance from respective RFMs (Figure 2B and Appendix A). All RFMs had a high prediction accuracy when exposed to new data, with model accuracies ranging from 93.1% to 99.9% accuracy (Figure 3 and Appendix A). Across all groups, the environmental variables salinity, temperature, and silicate were always the most important (consistent), while secondarily important variables were inconsistent across genera (Figure 2B and Appendix A). In addition to deviation among environmental variables, depth-specific variation also informed predicted niche space (Figure 2B). For example, *Cotylorhiza* and *Stomolophus* were both best predicted by deep (20 m) temperature data (Figure 2B). Underlying differences in associated environmental ranges further accentuated niche partitioning (Figure 2C–I).

Informed by RFMs (Appendix A), suitable habitat predictions typically reflected distribution data gleaned from iNaturalist (Figure 3 and Appendix A). Models ranged from predicting broad areas of suitable habitat (*Cassiopea*, *Lychnorhiza*, *Rhizostoma*, *Rhopilema*, *Stomolophus*) (Figure 3A,D and Appendix A) to more specific regions (*Catostylus*, *Cotylorhiza*, *Pseudorhiza*) (Figure 3B and Appendix A). In some cases, the most suitable habitats were predicted to be outside of the primary zone of occurrence data (*Eupilema*, *Lobonema*) (Figure 3C and Appendix A).

When compared to legacy distribution data, distribution models ranged in accuracy from 73.6% to 100% (Figure 3 and Appendix A). Accuracy was sometimes inflated when distribution models predicted extremely wide distributions (Appendix A). Legacy distribution data (latitude, longitude, and region) was from 180 publications manually collected between 25 January 2023 and 23 February 2023. The final legacy database consisted of 601 jellyfish records across the ten analyzed genera ranging in publication date from 1900 to January 2023 (Appendix A). Report data from the legacy database had high coordinal similarity to iNaturalist reports, but this does not necessarily translate to distribution models. All genus-specific random forest models and distribution prediction maps are found in FAppendix A.

## 4. Discussion

### 4.1. Inferring Ecological Niches

In environmental dimensions, most genera displayed overlapping niche spaces (Figure 2A). Only niches for *Stomolophus* and *Rhizostoma* were inferred as distinct from other genera (Figure 2A). Interestingly, both genera had high geographic overlap with one other genus: *Stomolophus* covaried with *Rhopilema* and *Rhizostoma* covaried with *Cotylorhiza*. Despite overlapping distributions, niche separation could contribute feasibly to a reduction in competition for resources between coexisting genera [58,59]. By contrast, genera with inferred overlapping niche space (Figure 2A) had distinct geographic distributions (Figure 1). Limited competition for resources, due to non-overlapping geographic distributions, may allow for niche redundancy, but this perceived redundancy could also be an artifact of GLM model biases. Dynamic interactions among an organism, biotic and abiotic factors, known as the ecological niche ‘space’, shape the suitability of an organism for maintaining that population in that space [60]. In this study, the data used to build our models had a relatively coarse spatial and temporal resolution, likely obscuring lower-resolution niche differences among genera. Therefore, while GLMs represent a well-understood statistical framework to model the relationships between predictor and response variables [56], data resolution likely obscured higher resolution niche differences between genera.

Higher resolution, machine learning RFMs suggested that either salinity, temperature, and/or silicate levels were the most important predictors for rhizostome jellyfish reports (Figure 2B). Salinity and temperature were the most influential predictors for most genera; nutrient concentrations (silicates, phosphates, and nitrates) were typically ancillary. A notable exception is *Lychnorhiza*, whose distribution was inferred to covary with high silicate, nitrate, and phosphate (Figure 2E,H,I). From understanding their habitat preference for high nutrient waters [61], this result seems initially intuitive. However, other jellyfishes that are also typically found in high-nutrient waters (e.g., *Cassiopea* [33]) did not show the same result (Figure 2B). Perhaps this indicates a heterotrophy-dependent distribution given plankton abundances covariance silicate, nitrate, and phosphate values, as may be expected given their extremely high feeding rates [62]. Conversely, *Cassiopea* medusae can better sustain themselves with autotrophy and are more influenced by temperature than nutrients (Figure 2B). *Lychnorhiza* are known to form large, harmful blooms [21]. Therefore, coastal environmental managers seeking to improve *Lychnorhiza* bloom prediction capabilities (e.g., season, location) may opt for tracking water nutrient levels to investigate possible correlations. This interpretation is simply a hypothesis and comes with its own set of limitations and biases. However, it is provided here as an example of how these models, together with valuable global-oriented databases, may act as the foundation for hypotheses that inform future experimental work, managerial strategies, and targeted regional studies.

In addition to using different abiotic predictors to inform ecological niches, this evidence allowed us to infer the occurrence [or suitability] of jellyfish taxa within a three-dimensional space, despite the two-dimensional data structure. This is a key component of our data and models, as marine organisms inhabit a three-dimensional environmental and ecological niche; therefore, modeling should include depth [63,64,65]. Given that the intrinsic nature of iNaturalist reports (photographs) primarily comprises reports of beached or shallow jellyfish lacking associated metadata (e.g., cooccurring taxa, substrate, or depth in situ), we are left to infer that jellyfish originated from nearby waters. While it was not possible to assign jellyfish reports to specific depths for model parameterization, the integration of environmental data across depth strata began to infer three-dimensional niche space. As hypothesized, some depths were more important to the models than others (e.g., 20 m vs. 0 m) (Figure 2B). For example, based on legacy distribution data, the cannonball jellyfish *Stomolophus* occurs between 10 m and 30 m depth [66]. Our RFMs accurately predicted this, heavily prioritizing the 20 m environmental data over the 0 m environmental data. This suggests that iNaturalist data can present consistent conclusions with legacy data despite limited ecological information. Given the limitations of the publicly available data generated by citizen participation efforts, model biases remain a challenge to solve. However, rather than disregarding this vast wealth of data, scientists can seize the opportunity to develop new tools that effectively predict, normalize, and mitigate biases. The work presented here is simply step one of facilitating responsible integration and analysis of citizen science data, ultimately leading towards effectively testing complex hypotheses and predicting marine ecological niches for a variety of taxa.

### 4.2. Predicting Geographic Distributions

Three forms of distributions were predicted from our citizen-science derived distribution models: (1) near-global neritic distributions (Figure 3A), (2) extremely tight distributions (Figure 3B), and (3) a small selection of predicted hotspots (Figure 3C). Our public data-derived distribution models may inform action-based management strategies to conserve limited populations or protect high-risk locations from future invasion [67,68]. Here, we will present *Lychnorhiza* data and models as a case study to lay out a framework for interpreting distribution models. *Lychnorhiza* has stable populations along the coast of Brazil up into Central America, a distribution corroborated herein by both legacy and iNaturalist data (Figure 3D). Interestingly, as discovered by Nogueira Jr. and Haddad [69], a single report of the genus had been reported on the Pacific coast of Mexico [70], but this report was not included in the species checklist [71]. We hypothesize that the exclusion of *Lychnorhiza* from the checklist suggests that there is not a stable population in Mexico and that the Cornelius and Silveira [70] report was an anomaly, possibly washed up from more established populations in Central America. Towards testing this hypothesis, we must consider whether there is a stable population in West Mexico, and, if so, we would expect two things: (1) iNaturalist reports documenting its occurrence or (2) a model-predicted, suitable habitat along the west coast of Mexico. Our iNaturalist data contained zero reports for the genus on the west coast of the Americas (Figure 1 and Figure 3). Conversely, many suitable habitats were predicted for *Lychnorhiza*, including the North Atlantic Ocean, Northwest Pacific Ocean, West Indian Ocean, Southeast Pacific Ocean, and even the west coast of Mexico, despite zero iNaturalist reports for *Lychnorhiza* in these regions (Figure 3D). This could be interpreted in one or more of the following ways: (1) *Lychnorhiza* has an extremely wide distribution and most populations have yet to be documented, (2) *Lychnorhiza* may be a candidate for future biological invasion, (3) our data and subsequently our models fail to include the predictor that influences the *Lychnorhiza* distribution. We cannot conclude whether the 1997 report from West Mexico [70] was an anomaly or not, but legacy data and distribution models suggest there could be a population there and in many other places around the world. For coastal managers, we would then recommend monitoring these regions of interest for *Lychnorhiza* populations to take inventory of current ranges and to detect future range expansions. Global models such as this are the first step to formulating a plan of action for ecosystem monitoring, management, and hypothesis-driven research.

In general, global synthesis studies on jellyfish struggle with non-systematic, temporally stochastic sampling and heavy geographic biases [27,28,29]. Online databases like iNaturalist struggle with similar issues [72]; however, proper application can lead to valuable insights and the development of new testable hypotheses and management strategies [22]. While niche and distribution models are informative and highlight the power of AI-driven identification of citizen-collected reports, the power is inhibited when data resolution is coarse taxonomically, environmentally, and geographically [72,73,74,75], such as was the case in this study. Taxonomically, we were limited to modeling at the genus level given the difficulty in confirming species-level identifications from photographs alone, considering the high number of cryptic species and morphological plasticity of Rhizostomeae [76]. This is one of the primary risks with using online databases to answer biogeographic questions [72]. In the future, incorporating computer vision models trained to make species-level predictions from jellyfish photographs based on expert identifications will permit finer-tuned models to predict ecological space.

Additionally, efforts are needed to improve geographic resolution of report data through encouraging increased data collection participation from currently underrepresented geographic regions. This would help resolve technical issues such as insufficient predictor variables (e.g., bathymetry) or participation bias (e.g., report abundance skewed towards Europe and North America). Depending on the intended use of niche and distribution models, ecological surveys, or further statistical analyses (e.g., geography-based report rarefaction) may be appropriate to address these shortcomings. For example, the *Cassiopea* model overpredicts known distributions due to a high concentration of reports in the Caribbean Sea in conjunction with a seemingly near-global neritic distribution. Additionally, bathymetry was not included as an abiotic environmental predictor and thus the model failed to recognize the distributional restriction of *Cassiopea* to shallow waters (Figure 3D). This prediction bias is present in most models, with the most suitable environment (dark areas) often appearing in the middle of the ocean (Figure 3D). Given the neritic habit of rhizostome jellyfishes, this is evidently incorrect. However, with proper interpretation, the most suitable environment may be applied to the region in which the darkest region resides (e.g., *Lobonema* may be predicted to occur on the east coast of Africa or the west coast of India; not necessarily the Central-West Indian Ocean (Figure 3C)). Overcoming these resolution issues requires expert knowledge on the biology of the organism and perhaps further analyses to adequately develop, analyze, and interpret databases and models.

### 4.3. Increasing Global Participation to Reduce Data Biases

At present, reports from iNaturalist heavily favor ‘Western’ societies, with most reports coming from the United States, Western Europe, or highly populated Australian coastlines (Figure 1A). This separation in participation is especially visible around the Mediterranean Sea, with high report density along the European coast contrasted with very few reports along the African coasts (Figure 1A). Other coastlines with high human densities and low participation include East China, Japan, India, Brazil, North and South American west coasts, West Africa, and Southeast Africa (Figure 1A). This bias is not necessarily due to a lack of scientific interest or participation, as China, Japan, India, and Brazil rank highly in natural sciences research output [77] and countries in Africa are experiencing strong scientific growth despite socioeconomic issues [78,79,80,81]. Therefore, the question is not ‘How can scientists facilitate citizens’ interest in science?’ but instead ‘How can scientists facilitate citizens’ participation in science?’ Implementing collaborations in target countries will involve flexible strategies, as international collaborations are directly related to geographic distance and socioeconomic conditions [82]. For example, citizen-science sampling strategies can range from basic to highly structured surveys, each with tradeoffs between data robustness and implementation difficulty [83]. Socioeconomically privileged countries that primarily collaborate domestically (within country) may be capable of highly structured surveys but require an intermediary expert to initiate and facilitate participation. Alternatively, internationally collaborative countries limited by socioeconomic factors may be easily accessible but require local public figures to generate participation in unstructured surveys through scientific communication. Ultimately, to increase participation through collaboration, we recommend targeting local experts, public figures, and hobbyists in underrepresented regions to implement projects locally.

## 5. Conclusions

The scientific community struggles with accessibility and inclusivity, particularly in marine biology [84,85,86]. In conjunction with systematic inequalities, many researchers lack access to adequate funding, field sites, and equipment, a problem slightly moderated by the development of public databases and tools. While these issues will persist for the foreseeable future, we believe that citizen science is a valuable tool for scientific research, providing both researchers and citizens with an opportunity to make meaningful scientific contributions and simultaneously reduce the height of the scientific paywall. Our findings represent just one of the many approaches that may be valuable for addressing the shortcomings and challenges that arise in this type of research. By contributing our curated rhizostome datasets and associated models, we encourage the scientific community to make use of this database while improving its quality and accuracy, managing volunteer motivation and retention, and addressing potential biases in participation. We propose the regionally targeted implementation of citizen science projects as it may help to overcome these challenges by improving data resolution and promoting global collaboration. This, in turn, will improve the power of these models to predict blooms, biological invasions, or hidden populations, which are the primary goals of jellyfish management. We must build partnerships with regionally relevant organizations, institutions, and communities. With this, we can increase the effectiveness and sustainability of citizen science projects by providing training and support for citizen scientists and leveraging technology (e.g., artificial intelligence) to enhance data collection and analysis. By recognizing and addressing these challenges, citizen science can play an increasingly important role in advancing scientific understanding amidst rapid change, promoting public engagement in scientific research, and breaking down systematic inequalities.

## Figures and Tables

**Figure 1 animals-13-01591-f001:**
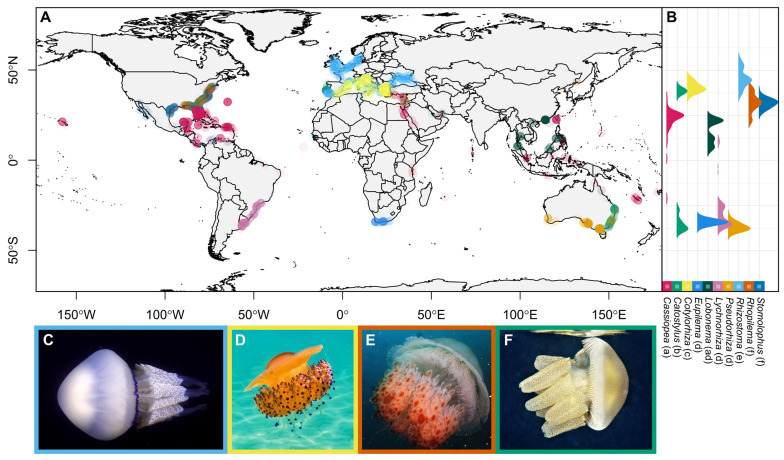
(**A**) iNaturalist reports of genera within the order Rhizostomeae with sufficient sampling determined by rarefied sampling coverage (SC > 95%) (Appendix A); circles represent individual reports. (**B**) Density curves along the y-axis visualize the report density across latitudes. Lowercase letters next to genus names indicate groups of statistically similar distributions based on post-hoc Dunn tests (*p* < 0.05). (**C**–**F**) Photos are examples of ‘Research Grade’ reports assigned the following taxon names on iNaturalist (inaturalist.org accessed on 3 November 2022): (**C**) barrel jelly *Rhizostoma pulmo* from Mauguio, France by Pascal GIRARD in 2022; (**D**) fried egg jelly *Cotylorhiza tuberculata* from Thessaly, Greece by marieta55 in 2021; (**E**) hizen kurage *Rhopilema hispidum* from Sai Kung, Hong Kong by Ryan Yue Wah Chan in 2021; (**F**) blubber jelly *Catostylus mosaicus* from NSW, Australia by John Sear in 2022 (photos under CC-BY-NC licenses).

**Figure 2 animals-13-01591-f002:**
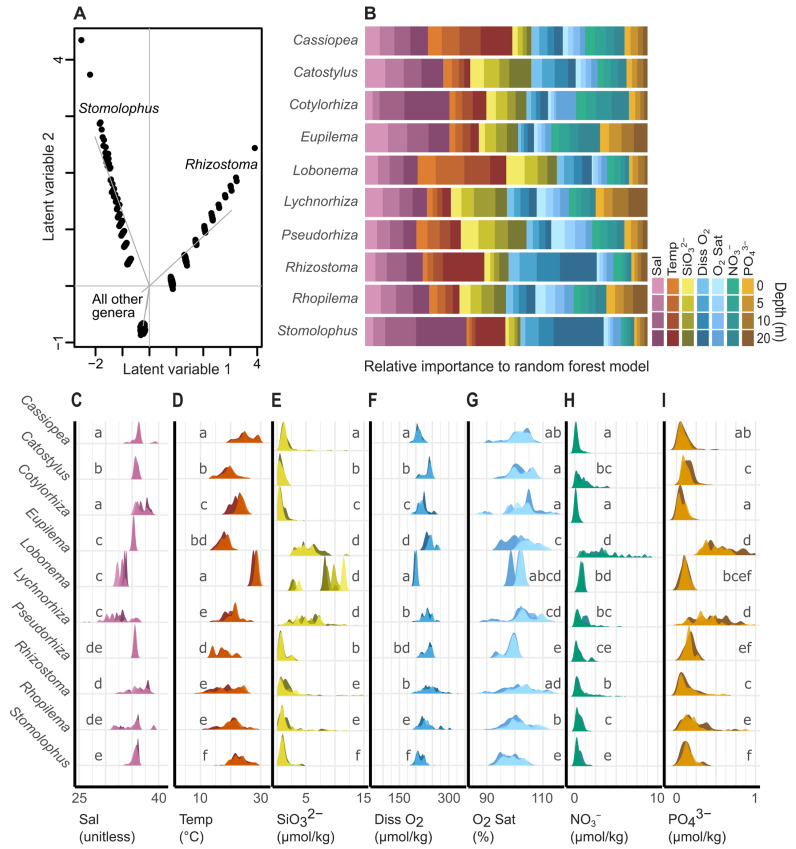
(**A**) Rhizostomeae niche space from generalized linear models visualized as a biplot with larger gaps between points indicating greater niche separation between genera. (**B**) Environmental variables (colors) and depths (shades) that best predict the occurrence of each genus as determined by random forest models. (**C**–**I**) Oceanic, environmental data used to build models in A and B: salinity, temperature, silicate, dissolved oxygen, oxygen saturation, nitrate, and phosphate. Lowercase letters in each row indicate statistically similar distributions based on post-hoc Dunn tests (*p* < 0.05).

**Figure 3 animals-13-01591-f003:**
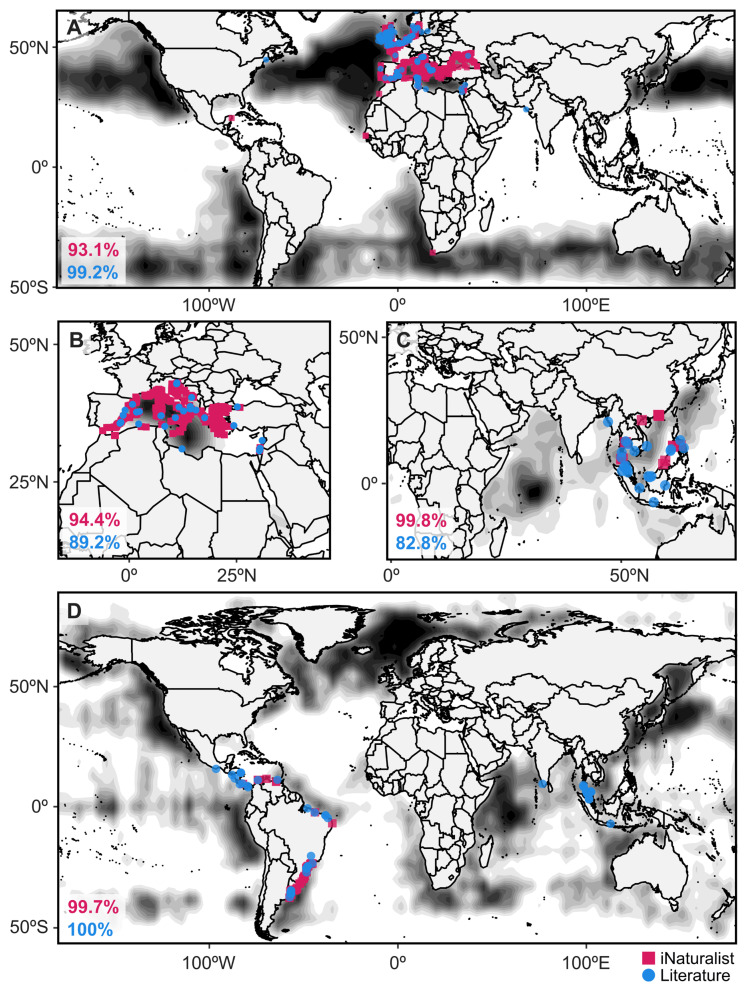
Predicted distributions of several Rhizostomeae genera based on RFMs built from iNaturalist reports (pink squares) and legacy data from the literature (blue circles). Dark density-based polygons indicate regions corresponding to suitable environments predicted by random forest models (see Figure 2B and Appendix A) to support the existence of each genus: (**A**) *Rhizostoma*, (**B**) *Cotylorhiza*, (**C**) *Lobonema*, and (**D**) *Lychnorhiza.* Percentages in pink font reflect RFM model accuracy built from iNaturalist data, while percentages in blue font are the distribution model accuracy based on legacy data (see Appendix A); both metrics can be used to infer confidence in the model.

## Data Availability

All data used in this research were from publicly available databases or previously published literature. Any original code and required databases have been permanently deposited on GitHub (https://doi.org/10.5281/zenodo.7897878).

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
