# Peer review of "Leveraging Public Data to Predict Global Niches and Distributions of Rhizostome Jellyfishes"

_animals, 2023, doi:10.3390/ani13101591_

Round 1

Reviewer 1 Report

General comments

 The manuscript entitled "Leveraging Public Data to Predict Global Niches and Distributions of Rhizostome Jellyfishes" by Colin J Anthony et al. reported about the data science on distributions and niches jellyfishes using citizen science data from iNaturalist and legacy data from published literatures. This attempt is very important because it can be applied not only to jellyfish research, but also to research on all aquatic organisms. As AI-based research develops in the future, it is expected to become more accurate. This will require global recognition of citizen science platforms and collaboration between researchers and the general public in each region. Therefore, this manuscript is worthwhile to be published in this journal.

However, I suspect there is quite a bias in selecting that species and environmental data, though it was a good selection Rhizostome jellyfish species as target animals despite the numerous species of jellyfish. As for environmental data, did you consider using the Google Earth Engine? Also, as for legacy data, if you are using the scientific name as it appears in the literature, that is a problem. It may be no different than citizen science information. It is possible that some of those references are still using the old scientific names or are misidentified, for example, Stomolophus nomurai is now Nemopilema nomurai, Most of Lobonema in the literatures you used may be Lobonemoides. Are these areas being corrected? To use data, It is very important to examine the species name. Just as there are many problems with species names for genes registered in the Genbank, it is very important to examine the species names in this study.

 Other queries and comments were as follows.

 Specific comments

L80: Muffett et al. 2021

Not listed in the reference section

L83: and.

Delete?

L88: Brotz et al. 2013

Not listed in the reference section

L88: Luo et al. 2020

Not listed in the reference section

L120: Garcia et al. 2018a

L121: Garcia et al. 2018b

Add “a, b” to the references in the bibliography.

L132: R Core Team 2021

L133: RStudio Team 2021

Not listed in the reference section

L118-135: About environmental data

Since Lychnorhiza and Acromitus are commonly found in mangroves and other river-influenced areas, is it possible to include habitat characteristics in the analysis? Also is it possible to include occurrence season?

L196: Lobonema

Is this Lobonema not Lobonemoides?

L205: Rhizostoma pulma

Rhizostoma pulmo

L206: GIRARD

Girard?

L207: eta55

Is it OK?

L207: Echizen-kurage

Hizen-kurage

L281-285: about Lychnorhiza distribution and nutrient concentration

As I have described before, it may be because Lychnorhiza are found in areas where there is river influence.

L322:Cornelius and Silveir 1997

L352: Brotz et al

L353:Luo et al

Not listed in the reference section

L478: Carvalho-Saucedo LF…

L491: Dalgard

L610: Purcell and Arai

Not cited in the text

Author Response

Dear Reviewer #1,

Thank you for the insightful comments. Your expertise on the topic is evident. I hope you will find our responses to your comments sufficient, and trust that we have fully integrated your comments to the best of our ability. I attempted to attach the revised manuscript to this response; however, it failed to upload to the online interface. If you are interested in seeing our revisions, please request a copy from the editor. Also note that in the text below, our responses to your comments are in bold font while your initial comments are in regular font.

Colin Anthony

Responses to Reviewer #1

The manuscript entitled "Leveraging Public Data to Predict Global Niches and Distributions of Rhizostome Jellyfishes" by Colin J Anthony et al. reported about the data science on distributions and niches jellyfishes using citizen science data from iNaturalist and legacy data from published literatures. This attempt is very important because it can be applied not only to jellyfish research, but also to research on all aquatic organisms. As AI-based research develops in the future, it is expected to become more accurate. This will require global recognition of citizen science platforms and collaboration between researchers and the general public in each region. Therefore, this manuscript is worthwhile to be published in this journal.

However, I suspect there is quite a bias in selecting that species and environmental data, though it was a good selection Rhizostome jellyfish species as target animals despite the numerous species of jellyfish.

As for environmental data, did you consider using the Google Earth Engine?

Thank you for the suggestion. Google Earth has a nice set of data, especially for the terrestrial ecosystem. The data google earth uses for seawater temperature and other metrics is derived from NOAA, therefore we opted for gathering the data directly from the NOAA database. Chlorophyll data was also considered given its relationship to phytoplankton; however, to consider ecological niches within a three-dimensional space, we only used independent variables with both coordinate-based data and depth data. We have added a sentence in the discussion calling for the possible incorporation of chlorophyll if the experimental design allows for it. We will keep the Google Earth Engine in mind for future projects.

Also, as for legacy data, if you are using the scientific name as it appears in the literature, that is a problem. It may be no different than citizen science information. It is possible that some of those references are still using the old scientific names or are misidentified, for example, Stomolophus nomurai is now Nemopilema nomurai, Most of Lobonema in the literatures you used may be Lobonemoides. Are these areas being corrected? To use data, It is very important to examine the species name. Just as there are many problems with species names for genes registered in the Genbank, it is very important to examine the species names in this study.

Thank you very much for pointing this out. We have worked through each species in the ‘legacy’ database to make sure it is still within the correct genus. We also converted species to the correct species if possible. If the genus it belongs to now (e.g Nemopilema) was not within the 10 genera included in this study, it was removed from the dataset. We have also added a brief explanation to the methods section explaining this process (Lines 137-142). Thank you again for the insightful comment.

Here are the exact changes we made in response to this comment:

  • Stomolophus was checked, and the only issue we discovered was the reassignment of Stomolophus nomurai to Nemopilema. These reports have been removed from our database.
  • In reference to Lobonema, Lobonema mayeri was converted to Lobonema smithii, as indicated by the WORMS taxonomy database. We checked all other Lobonema identifications in our legacy database and found no reason to believe, given the current state of the literature on the group, that they were misidentifications. From what I can tell the two genera, have similar distributions within China, Indonesia, and the Philippines. Future work may reveal currently hidden functional diversity, and in turn misidentifications. However, right now, we believe the data we present here is reliable and the best version of data. We truly appreciate your comment, this is something we initially overlooked, but have now resolved to the best of our ability.
  • In Rhizostoma¸ we found no issues. Authors had already considered taxonomic revisions upon publication and converted names accordingly (e.g. Rhizostoma hispidum -> Rhopilema hispidum).
  • In Rhopilema, R. esculenta was converted to esculentum.
  • No issues were found within Cotylorhiza, Catostylus, Rhopilema, Lychnorhiza, Pseudorhiza, or Eupilema.

All figures and databases have been updated to adhere to these changes, including supplemental files and the GitHub repository.

 Other queries and comments were as follows.

 Specific comments

L80: Muffett et al. 2021

Not listed in the reference section

Thank you for the catch; Muffett et al. 2021 has been added to the references section.

L83: and.

Delete?

 Thank you. This has been fixed.

L88: Brotz et al. 2013

Not listed in the reference section

 It was meant to be Brotz et al. 2012. This has been added to the references.

L88: Luo et al. 2020

Not listed in the reference section

 Luo et al. 2020 has been added to the reference section.

L120: Garcia et al. 2018a

L121: Garcia et al. 2018b

Add “a, b” to the references in the bibliography.

 This has been resolved. Thank you.

L132: R Core Team 2021

L133: RStudio Team 2021

Not listed in the reference section

These have been added. Thank you for the comment.

L118-135: About environmental data

Since Lychnorhiza and Acromitus are commonly found in mangroves and other river-influenced areas, is it possible to include habitat characteristics in the analysis? Also is it possible to include occurrence season?

Thank you for this comment. As you mention later in this review, the data does show signs of being able to identify habitat characteristics like this based on the Random Forest Models. Also, all environmental data and jellyfish reports were paired within the same month. Therefore, all models automatically incorporate seasonality related influences into the distributional predictions. We have made this point clearer in the methods section now (Line 125-128). Thank you for contributing your expertise, your point strengthens our discussion of the models, which is better responded to later in this document.

L196: Lobonema

Is this Lobonema not Lobonemoides?

Yes, this is in reference to Lobonema as we found no evidence to believe there were misidentifications in our literature-derived database (discussed above).

L205: Rhizostoma pulma

Rhizostoma pulmo

 This has been fixed. Thank you.

L206: GIRARD

Girard?

 This is a username on inaturalist.org; therefore, it must be formatted in this way.

L207: eta55

Is it OK?

 This is okay. It is the correct username from inaturalist.org, referencing the person that took the photo.

L207: Echizen-kurage

Hizen-kurage

Thank you for the correction! I did not notice this mistake and am deeply grateful for this comment.

L281-285: about Lychnorhiza distribution and nutrient concentration

As I have described before, it may be because Lychnorhiza are found in areas where there is river influence.

 Thank you for the comment! We have extended this discussion point based on this recommendation. (Lines 291-302)

L322:Cornelius and Silveir 1997

This has been added to the references list. We also added a citation for the original article that discovered this pattern. (Line 330)

L352: Brotz et al

This has been added to the references list.

L353:Luo et al

Not listed in the reference section

This has been added to the references list.

L478: Carvalho-Saucedo LF…

L491: Dalgard

L610: Purcell and Arai

Not cited in the text

Thank you for noticing this. These citations have been removed from the references list.

Reviewer 2 Report

The paper “Leveraging Public Data to Predict Global Niches and Distributions of Rhizostome Jellyfishes” is an excellent step in the ongoing task of incorporating citizen science data into scientific analyses. The paper is well written, comprehensive, and provides a fascinating analysis of global jellyfish distributions. The authors do an excellent job of framing their paper, describing their analyses and their limitations, and providing clear next steps and use for their dataset. The study provides a valuable dataset and methodology for studying the biogeography and niche partitioning of these ecologically and economically important species. I particularly commend the authors for providing excellent supplementary information, code, and datasets through Github. I have no suggestions beyond the two minor comments below.

Line 33 – Do you mean, “there are few jellyfish long-term, high-sample datasets…”?

Line 114-115 – How was raw report data converted to abundance data?

Author Response

Dear Reviewer #2,

We are glad you found our manuscript insightful. I hope you will find our responses to your comments sufficient, and trust that we have fully integrated your comments to the best of our ability. I attempted to attach the revised manuscript to this response; however, it failed to upload to the online interface. If you are interested in seeing our revisions, please request a copy from the editor. Also note that in the text below, our responses to your comments are in bold font while your initial comments are in regular font.

Colin Anthony

Reviewer #2

The paper “Leveraging Public Data to Predict Global Niches and Distributions of Rhizostome Jellyfishes” is an excellent step in the ongoing task of incorporating citizen science data into scientific analyses. The paper is well written, comprehensive, and provides a fascinating analysis of global jellyfish distributions. The authors do an excellent job of framing their paper, describing their analyses and their limitations, and providing clear next steps and use for their dataset. The study provides a valuable dataset and methodology for studying the biogeography and niche partitioning of these ecologically and economically important species. I particularly commend the authors for providing excellent supplementary information, code, and datasets through Github. I have no suggestions beyond the two minor comments below.

Thank you for the positive review. We are also excited about the data accessibility associated with this article. We hope the scientific community will find it valuable.

Line 33 – Do you mean, “there are few jellyfish long-term, high-sample datasets…”?

Thank you for the correction. It has been resolved (Line 33).

Line 114-115 – How was raw report data converted to abundance data?

Thank you for the comment. We have added more detail to this sentence to clarify our methodology (Line 115-117).

Reviewer 3 Report

Simple Summary.

In the first sentence (Line 13) the authors use the word ‘prevent’ which creates an odd sense of the work presented.  If iNaturalist does not prevent resources managers from using these data, what is the point of the paper?

Abstract:

Lines 32-33.  [Despite …difficult].  This sentence does not make sense (ibid the comment from Simple Summary).  Are you implying that despite robust, long-term, high-sample data sets, the management community can’t use these data sets?

Introduction.

L 52-54.  First sentence is confusing and awkward.  This needs to be rewritten in a manner that it easily understood upon first reading.

L 81-82.  Rajagopal reference not in References section.  However, the other 2 references regarding Purcell are not related to sustainable energy operation.  Perhaps that is because ‘sustainable’ implies renewable energy resources and in the marine realm this is currently restricted to a large degree with Wind power.  There is no evidence of how jelly blooms would impact wind energy.

L 83.  Dropped sentence? After reference, “), and.”  Not sure if something is missing, but this should have been picked up prior to submission.

Methods and Results

Editing clean up needed

Discussion

I recognize that this is one-step toward utilizing citizen data to generate larger scale patterns of abundance and then link them to other factors and that there are a lot of sampling bias problems that exist as well as statistical framework challenges.  Ultimately, for this paper this can be overlooked, but I think you need to add a little more uncertainty or caveats related to your predictions.

However, one thing that I think you need to discuss in a bit more realistic terms is how this work helps the management community.  You set this up in the Summary and Abstract, but the link to the management decision making process does not really occur.  What are ‘managers’ supposed to do with your model results? 

Is this about developing warning systems based on environmental, depth, etc… data that blooms may occur? 

Do the taxa you are looking at pose significant threats and if so, why/how?

Can you realistically manage jellyfish medusa without understanding the real problem which is the polyp stage that generates the adults?

Details

Line 23.  Last word (to) should be replaced with ‘of’.

Line 112.  Replace ‘is’ with ‘are’ (plural clause)

Line 114.  I am old school, ‘Data’ is a plural noun, so the verb should be were.

Line 213.  Italics of Rhizostoma

Line 214.  Italics Cotylorhiza

Line 520-521.  Incomplete reference

Author Response

Dear Reviewer #3,

Thank you for spending the time to review our manuscript. Your insight was helpful and improved the manuscripts narrative and clarity. I hope you will find our responses to your comments sufficient, and trust that we have fully integrated your comments to the best of our ability. I attempted to attach the revised manuscript to this response; however, it failed to upload to the online interface. If you are interested in seeing our revisions, please request a copy from the editor. Also note that in the text below, our responses to your comments are in bold font while your initial comments are in regular font.

Colin Anthony

Reviewer #3

Simple Summary.

In the first sentence (Line 13) the authors use the word ‘prevent’ which creates an odd sense of the work presented.  If iNaturalist does not prevent resources managers from using these data, what is the point of the paper?

 Thank you for the comment. We have adjusted the sentence to be more direct.

Abstract:

Lines 32-33.  [Despite …difficult].  This sentence does not make sense (ibid the comment from Simple Summary).  Are you implying that despite robust, long-term, high-sample data sets, the management community can’t use these data sets?

 The sentence has been modified to specify that there are few datasets, which makes management difficult.

Introduction.

L 52-54.  First sentence is confusing and awkward.  This needs to be rewritten in a manner that it easily understood upon first reading.

The sentence has been revised to improve clarity. Thank you for the comment.

L 81-82.  Rajagopal reference not in References section.  However, the other 2 references regarding Purcell are not related to sustainable energy operation.  Perhaps that is because ‘sustainable’ implies renewable energy resources and in the marine realm this is currently restricted to a large degree with Wind power.  There is no evidence of how jelly blooms would impact wind energy.

 Thank you for the comment. We have opted to remove this statement, as it was not well enough supported and did not contribute to the narrative.

L 83.  Dropped sentence? After reference, “), and.”  Not sure if something is missing, but this should have been picked up prior to submission.

 This has been resolved. Thank you for pointing out our oversight.

Methods and Results

Editing clean up needed

 Thank you for the comment. All authors have passed through the methods and results to resolve any grammatical or conceptual issues. We hope you will find the revision satisfactory.

Discussion

I recognize that this is one-step toward utilizing citizen data to generate larger scale patterns of abundance and then link them to other factors and that there are a lot of sampling bias problems that exist as well as statistical framework challenges.  Ultimately, for this paper this can be overlooked, but I think you need to add a little more uncertainty or caveats related to your predictions.

Yes, we agree with you. In this manuscript, we did not attempt to resolve jellyfish biology, but instead highlighted the powerful potential associated with publicly available data, as well as discussed its biases and limitations. We have worked through the discussion to make sure the biases are more readily acknowledged. Here are all of the lines, which acknowledge sampling bias or resolution issues (Lines 284-289; 307-311; 328-334; 375-409; 411-413; 446-449), of which includes a two-paragraph section specifically discussing resolution limitations and biases of our data and analysis (375-409).

However, one thing that I think you need to discuss in a bit more realistic terms is how this work helps the management community.  You set this up in the Summary and Abstract, but the link to the management decision making process does not really occur.  What are ‘managers’ supposed to do with your model results? 

We appreciate your comment, and agree with you. We added an example for how managers may use this data to predict blooms (Lines 300-309) and then added a sentence in the conclusion to mention the relevance for managers (451-453). We also mention in section 2.4 how this data can be used for managers to evaluate the risk of invasion or range expansion (338-341), followed by using Lychnorhiza as a case study for interpreting our distribution models (Lines 341-367). We hope the application of this model to managers and other scientists has become more conspicuous after this revision.

Is this about developing warning systems based on environmental, depth, etc… data that blooms may occur? 

This is certainly one way the data could be used. We discuss it briefly in lines 300-309, and then added a conclusion sentence to help make the relevance of this methodology to managers and other researchers more conspicuous.

Do the taxa you are looking at pose significant threats and if so, why/how?

Yes, some taxa pose significant threats, as mentioned about Lychnorhiza (Line 300 – 309), but others are desired products (e.g. for food), as discussed in the introduction. Given the use of 10 different genera, we did not find it valuable to discuss the economic relevance of each individual species, unless directly relevant for the discussion. Instead, we focused on highlighting the broad relevance of researching the order with this type of data structure.

We hope scientists will be able to take the research design discussed here and apply it jellyfish or other marine organisms that are relevant for them.

Can you realistically manage jellyfish medusa without understanding the real problem which is the polyp stage that generates the adults?

We believe you can, as medusa are a byproduct of the polyp stage. Understanding the seasonality of medusae inherently provides information as to the seasonality of the polyp. Of course other research, such as settlement substrate preferences is important for improving our predictive capacity of jellyfish blooms, but understanding the triggers of strobilation and retrospectively the likelihood of medusa occurrence (as done here) is a first step. This is the direction our paper takes. We believe this direct relationship between polyp and medusa is sufficiently covered in the introduction (Line 72-80).

Details

Line 23.  Last word (to) should be replaced with ‘of’.

Thank you for this comment. We appreciate you noticing this. It has been fixed.

Line 112.  Replace ‘is’ with ‘are’ (plural clause)

We fixed this. Thank you.

Line 114.  I am old school, ‘Data’ is a plural noun, so the verb should be were.

Thank you. It has been fixed.

Line 213.  Italics of Rhizostoma

Thank you. This has been fixed.

Line 214.  Italics Cotylorhiza

Thank you. This has been fixed.

Line 520-521.  Incomplete reference

Lawley et al. 2022 has been completed. Thank you for catching that.

Round 2

Reviewer 1 Report

The manuscript was greatly revised.

Data science research that combines citizen science on the internet and past legacy data will be important for jellyfish research in the future.

As for Lychnorhiza, I would like you to analyze them at the species level. You had better try this before acceptance , this manuscript would be great paper.

Author Response

"The manuscript was greatly revised.

Data science research that combines citizen science on the internet and past legacy data will be important for jellyfish research in the future.

As for Lychnorhiza, I would like you to analyze them at the species level. You had better try this before acceptance, this manuscript would be great paper."

While we appreciate your response, and are glad that you find our paper beneficial, the underlying data doesn't have the taxonomic resolution to conduct species-level analyses, as discussed in our manuscript. While during our previous revision, we did our best to validate the identifications of reports, our contribution focuses on the niche partitioning and distributions of rhizostome jellyfishes at the genus level in order to avoid the high rates of misidentifications at the species level that are inherent in citizen science data.

Our contribution lays the groundwork for the types of analyses you suggest. We hope species-level analyses will become feasible with the continued development of publicly available data. We believe that our manuscript has the potential to spur the continued development of publicly available databases, and in-turn the resolution limitations that we discuss in our manuscript.

Taxonomic resolution of iNaturalist-based models is only reliable at the genus-level until future researchers develop strategies to reliably improve the resolution of publicly available databases. Given these considerations, we hope you understand why we cannot conduct the requested analysis.